# Characteristics, treatment outcomes and factors associated with death among patients with Visceral Leishmaniasis, Uganda, 2019–2024

Benigna Gabriela Namara[1]*, Ivan Ankunda[2], Richard Migisha[1], Benon Kwesiga[1], Lilian Bulage[1], Sandra Nabatanzi[3], Alex Riolexus Ario[1], Alfred Mubangizi[2], Daniel Kadobera[3]

1 Uganda Public Health Fellowship Program, Uganda National Institute of Public Health, Kampala, Uganda, 2 Vector Borne and Neglected Tropical Diseases Division, Ministry of Health, Kampala, Uganda, 3 United States of America Centers for Disease Control and Prevention, Kampala, Uganda

* benignamara@uniph.go.ug

## Abstract

### Background

Visceral leishmaniasis (VL), a neglected tropical disease (NTD)continues to affect several countries worldwide, including Uganda, where it remains a significant public health concern in the Karamoja Region. This region borders Kenya, where VL is endemic. Globally and within East Africa, VL persists due to a combination of ecological suitability for sandfly vectors, chronic underdiagnosis, limited access to care in remote and pastoralist communities, high levels of malnutrition and poverty, and cross-border population movement that sustains transmission. The World Health Organization (WHO) targets to eliminate VL as a public health problem by reducing case fatality to <1%, but the current burden of VL is unknown. We described VL patients, their treatment outcomes, and identified factors associated with death in Uganda, from 2019–2024, to check progress towards meeting the country's targets.

### Methods

We conducted a retrospective observational review of patient records from 2019–2024 at the main VL treatment center in Amudat District, Uganda, abstracting socio-demographic, clinical, treatment, and outcome dataWe used logistic regression to determine factors associated with death.

### Results

Among 972 patients, 670 (69%) were male and 742 (76%) were age ≤ 18 years. Three hundred and seventy-three (38%) were from Kenya, while most, 434/599 (72%) Ugandan patients were from Moroto District. The highest number of cases (322) was recorded in 2022, with Ugandans making up 80% of all patients that

purpose. The work is made available under the Creative Commons CC0 public domain dedication.

**Data availability statement:** The data used in this submission has been attached as a supplementary file.

**Funding:** This study was supported by the President's Emergency Plan for AIDS Relief through the United States Centers for Disease Control and Prevention Cooperative Agreement number GH001353-01 to BGN, through Makerere University School of Public Health to the Uganda Public Health Fellowship Program, Ministry of Health. The contents of this manuscript are solely the responsibility of the authors and do not necessarily represent the official views of the US Centers for Disease Control and Prevention and the Department of Health and Human Services, Makerere University School of Public Health, or the Uganda Ministry of Health. However, this funding does not cover publication costs.The funders participated in the review process and provided editorial feedback on the manuscript.

**Competing interests:** The authors have declared that no competing interests exist.

year(259/322), unlike previous years (2019–2021) when Kenyan patients predominated. There was no identifiable seasonal pattern/variation in the number of cases diagnosed. The commonest symptoms were fever (98%), night sweats (77%), and abdominal swelling (72%). The average duration of sickness was 2.6 months (standard deviation (SD)=0.3 months). Severe anemia was common (512/972; 53%), and among the patients tested for co-infections, 175/969 (18%) were co-infected with malaria and 185/593 (31%) with Human Immunodeficiency Virus (HIV). For most patients, 898 (92%), this was their index episode of VL. Almost all patients [957 (98%)] were cured. and most [743 (76%)] patients were treated with the 1st- line regimen. The case fatality rate (CFR) declined from 2% in 2020 and 2021 to <1% in 2023 and 2024. Being HIV positive was associated with death (Adjusted odds ratios (AOR) 10, 95% Confidence Intervals (CI) 2.2-50, p = 0.003).

## Conclusion

This study indicates progress towards the elimination of VL while highlighting the significance of cross-border transmission and the importance of screening/treatment of co-infections, especially HIV.

## Author summary

Visceral leishmaniasis (VL), also known as kala-azar, remains a significant but under-researched public health concern in Uganda, particularly in the semi-arid, cross-border Karamoja region. This retrospective study analyzed individual-level data from 972 patients treated for VL at Amudat Hospital between 2019 and 2024 to characterize patient profiles, treatment outcomes, and factors associated with mortality. Most patients were male children, with over one-third originating from neighboring Kenya, highlighting the cross-border nature of VL transmission. The majority presented with typical VL symptoms—fever, night sweats, abdominal swelling, and anemia—and nearly all were diagnosed using rapid diagnostic tests. HIV co-infection was common (30%) and strongly associated with death, making it the only significant predictor of mortality in multivariate analysis. The overall case fatality rate was low (1%), with most patients completing treatment and being cured, indicating substantial progress toward VL elimination targets. However, findings underscore the need for continued surveillance, improved management of co-infections, and coordinated cross-border interventions to interrupt transmission and achieve national and regional VL elimination goals.

## Introduction

Visceral leishmaniasis (VL), also known as kala-azar, is a neglected tropical disease caused by a protozoan parasite of the genus *Leishmania.* It is spread from human to

human or from animal to human through the bite of an infected sand fly of the genus Phlebotomus [1–5]. The main cause of VL in Uganda is *Leishmania donovani* and humans are considered the main reservoir for this parasite [6–9]. Sandflies are attracted to crowded housing because it is easier to bite people and feed on their blood. In addition, human behaviour such as sleeping outside or on the ground may increase the risk of being bitten by a sand-fly [10]. The typical clinical features of the disease include fever, weight loss, hepatosplenomegaly, and anemia [2,11].

The disease remains a significant public health problem in Uganda and is one of the Neglected Tropical Diseases (NTDs) that afflicts mainly the Karamoja region, inhabited mainly by pastoralists, where it was first detected in 1951 [12]. This region borders the Turkana and West Pokot regions of Kenya, also pastoralist regions, where VL is highly endemic. Despite a number of interventions, VL has remained endemic in the Karamoja region, and it is thought that cross-border migration between Kenya (where VL is also endemic) and Uganda has contributed to its continuous transmission [13,14].

A historical review of VL published in 2014 indicated that the prevalence was higher among males than females and among the 5–14 year age group, although the current burden is unknown [15]. A review of data in 2022, on reported cases from health facilities in Uganda reported an average annual incidence of 4 cases per 1,000,000 nationally but a higher burden (139–477 cases per 1,000,000) in Amudat district in the Karamoja region, equivalent to an average of 11–40 cases per month, of which only 11% were laboratory confirmed [16].

Reported risk factors for VL include malnutrition, immunosuppression due mainly to HIV infection and genetic susceptibility, low socio-economic status, and treating livestock with insecticide, which is common in the Karamoja region [15]. On the other hand, sleeping near animals, owning a mosquito bed net, and knowledge about VL were associated with reduced risks for contracting VL [17]. The risk factors identified for in-hospital deaths were age < 6 years and age > 15 years, concomitant tuberculosis, and liver disease [18]. A recent (2022) study using routinely collected surveillance data from the District Health Information System version 2 (DHIS2) reported an annual case fatality rate (CFR) of 5% [16]. However, since 2014, there has been no new information on possible risk factors for death, disease outcomes or treatment outcomes. This information is useful for guiding or improving programmatic interventions aimed at the elimination of the disease.

In Uganda, treatment for leishmaniasis is available mainly in three specific treatment centers in three Karamoja districts—Amudat, which is the main and oldest, Moroto, which has been functional for less than 2 years, and Napak for less than one year [19]. Treatment of this disease consists of a first-line 17-day drug regimen of sodium stibogluconate (S.S.G) + paromomycin (PM) administered intramuscularly daily or a second-line 14-day regimen of liposomal Amphotericin B administered intravenously, for special categories including pregnant women, VL-HIV co-infection, and failure of first-line. These drugs can have severe side effects, which may affect adherence and treatment outcomes. Without treatment, the disease is invariably fatal [13]. Treatment outcomes for VL in Uganda are largely undocumented.

All patients meeting the clinical case definition for VL need to be confirmed serologically with Rapid Diagnostic (Blood Dipstick) Tests (RDTs) or parasitologically with a spleen or bone marrow biopsy followed by microscopic identification of amastigotes on stained slides [20]. Diagnosis of VL in Uganda relies mainly on the RDTs which allows for an earlier diagnosis and treatment, and therefore an improved prognosis, whereas parasitological diagnosis in not common.

Uganda targets to eliminate VL as a public health problem by 2030, with 85% of the cases detected, 95% treated, and a < 1% CFR [21]. This underscores the need to reassess the status of VL in Uganda to date, characterize the patients, and evaluate treatment outcomes, and the factors associated. This would inform the development of targeted interventions—such as enhanced surveillance and early treatment in high-risk districts—towards accelerating achievement of the country's targets. The most recent study on VL in Uganda (2022) only described the spatial and temporal distribution of VL in Uganda using aggregated data from the DHIS2. This study did not fully characterize these patients or determine their treatment outcomes and associated factors [16].

Efforts to eliminate VL as a public health problem have been undertaken in several high-burden settings, offering important lessons for Uganda. In East Africa, the regional VL elimination strategy led by Kenya, Ethiopia, Sudan, South Sudan,

and Uganda emphasizes early case detection, expanded access to effective treatment regimens, and strengthening cross-border coordination to interrupt transmission [22]. Similarly, the VL elimination initiative in the Indian subcontinent (India, Nepal, and Bangladesh) achieved dramatic reductions in incidence through sustained vector control, decentralized diagnosis with rapid tests, community-based surveillance, and investment in strong referral systems [23,24].

These programs demonstrate that elimination is feasible when supported by systematic surveillance, early treatment, and coordinated regional action, all of which are highly relevant to Uganda's elimination targets."

Despite decades of VL control efforts in Uganda, the true clinical burden, treatment outcomes, and mortality risk factors remain poorly documented, particularly in the high-transmission Karamoja region. Existing national data rely almost entirely on aggregated DHIS2 reports, which are limited by under-diagnosis, variable reporting quality, and lack of patient-level detail necessary to guide elimination efforts. Moreover, the most recent published analyses date back to 2014 and 2022, neither of which characterized individual patients nor examined clinical outcomes or predictors of death, leaving critical evidence gaps in a context where Uganda aims to reduce VL case fatality to <1% by 2030.

Given the region's unique epidemiologic profile—including intense cross-border movement with highly endemic Kenyan districts, high malnutrition rates, and the increasing importance of HIV co-infection—current, granular data are urgently needed to assess progress toward national targets and identify modifiable factors that influence patient outcomes. By analyzing individual-level data from the country's main treatment center over a five-year period, this study fills a key gap in understanding who is affected, how patients are currently faring under routine program conditions, and what factors continue to drive mortality, thereby providing actionable evidence to strengthen surveillance, case management, and cross-border VL control strategies.

We described VL patients, their treatment outcomes, and identified the factors associated with death in Uganda during 2019–2024 using individual-level patient data from the main treatment center, Amudat Hospital, in Amudat District in Karamoja Region.

## Methods

### Study design, setting, and data source

We conducted a retrospective observational review of patient records from 2019–2024 at the main VL treatment center (Amudat Hospital) in Amudat District, Uganda.The study period was selected to capture the most recent five years of patient data, providing a contemporary view of VL trends while remaining feasible within the available time and resources for data collection and analysis. Amudat hospital was selected because it is the main VL treatment center and has been operational since 2000, while the newer centers in Moroto and Napak have only been operational since 2023.Consequently, the study reflects a convenience sample of patients presenting at Amudat rather than a nationally representative population.

The hospital is located in Amudat district in the Karamoja region, which lies in northeastern Uganda (Fig 1). The district is bordered by Nakapiripirit District to the North, Moroto District to the Northwest, and Kenya to the East. The population is predominantly rural, and the people are largely pastoralists who travel to and from the neighbouring VL endemic regions in Kenya. The region is semi-arid with long dry seasons in which sandflies thrive. The predominant sand-fly vector for VL in this region is *Phlebotomus martini* [3].

Because the people in this region are pastoralists, they tend to keep large numbers of livestock within their homesteads, which act as feeding grounds for sandflies and increase the risk of human exposure to sandflies. In addition, Karamoja has one of the highest rates of malnutrition in Uganda, particularly among children, which compromises the immune system, making them more vulnerable to VL [25]. Conversely, the region has an HIV prevalence of 2.1%, the lowest in the country [26]. Surveillance of VL is passive, based on health facility data on patients diagnosed with VL, data is aggregated, entered, and reported monthly into DHIS2. Individual-level data is kept in the patient files at the facility.

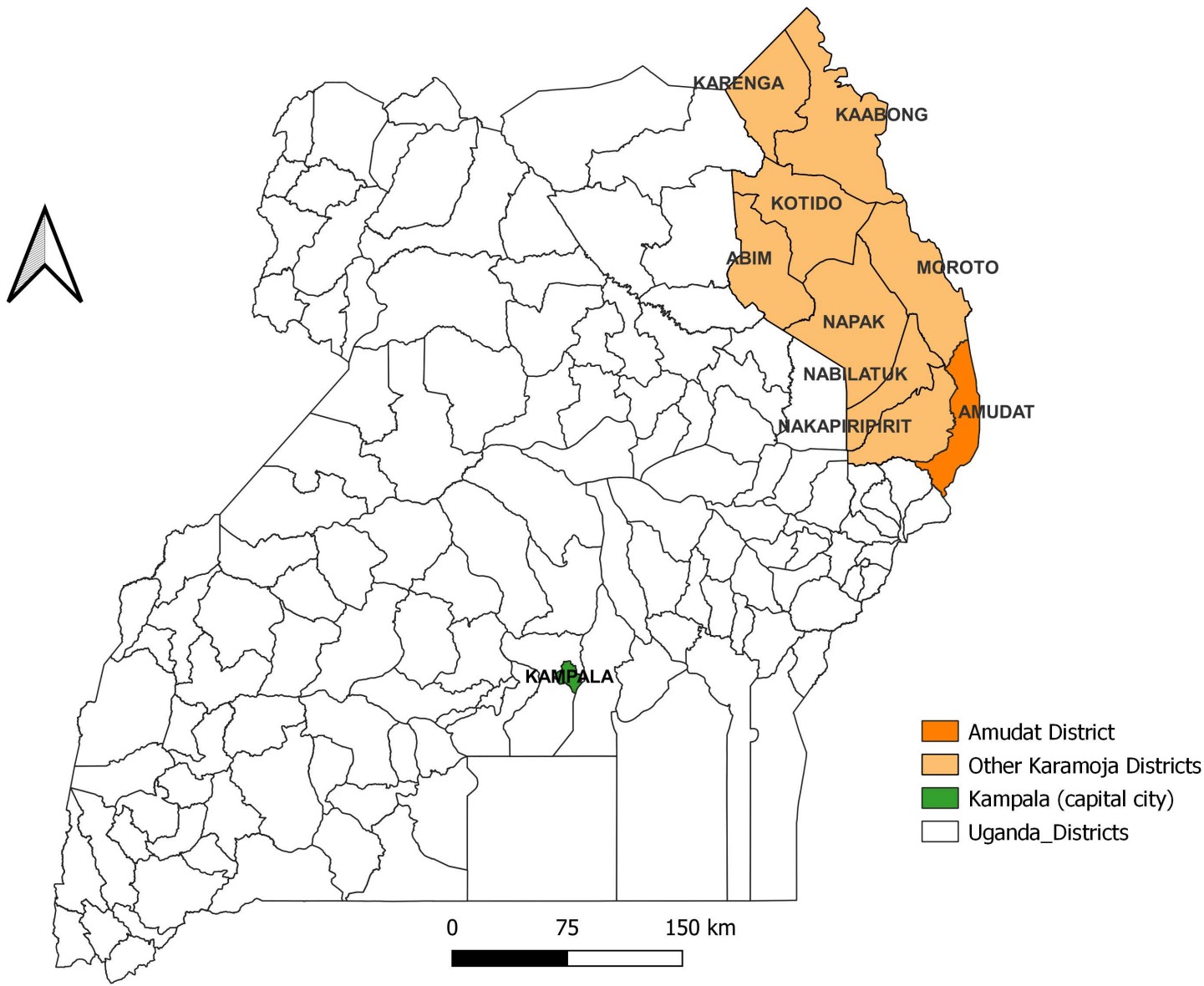

**Fig 1. Amudat District, Karamoja, Uganda (Map created in QGIS 3.10.2 using publicly available administrative boundary shapefiles from the Uganda Bureau of Statistics (UBOS), accessed via the Humanitarian Data Exchange: https://data.humdata.org/dataset/cod-ab-uga).** Map showing the geographical location of Amudat District in northeastern Uganda, where Amudat Hospital—the main visceral leishmaniasis (VL) treatment center during the study period—is situated.

## Study variables and data abstraction

Using a Kobocollect tool, we abstracted data on patient counts, socio-demographic characteristics, clinical signs and symptoms, date of diagnosis, date of treatment start, and completion/termination, laboratory results, co-morbidities, treatment regimen, length of admission, and disease outcomes from the patient files. Because some NTDs are targeted for elimination, changes in absolute patient numbers are usually used as a measure of the burden of disease and progress towards elimination [27].

## Data analysis

We analyzed the data using Microsoft Excel and Stata version 12.0. We calculated proportions for categorical variables, including patient characteristics such as sex, residence, symptoms, co-morbidities, treatment, and treatment outcomes; and summary statistics for continuous variables such as age and duration of illness using means and standard deviation (SD). We assessed for seasonal variation by comparing the numbers of VL cases in the months of the rainy season with those in the dry season. We employed logistic regression models to identify factors associated with death. The multivariable model was fitted using a bidirectional stepwise elimination approach. Only predictive variables with a significance level of $p \leq 0.2$ at bivariate analysis were included in the model after performing chi-square tests for independence. To assess the model's fit, we used the Hosmer-Lemeshow goodness-of-fit test. Odds ratios (ORs) and their corresponding 95% confidence intervals (CIs) were examined to quantify the strength and direction of association between predictors and death. Cases with missing values for any of the predictor variables were excluded from the analysis.

## Results

### Demographic characteristics of Visceral Leishmaniasis patients, Uganda, 2019–2024

A total of 972 patient records were reviewed for the period 2019–2024; 670 (69%) were male, the mean (standard deviation[SD]) age was 14(SD 2) years. The majority (61.6%) of VL patients were Ugandan, but 373 (38%) came across the border in Kenya (the majority (38%) of whom lived in the Turkana region). Most (72.5%, n = 434) of the Ugandan patients were from Moroto district. The highest number of cases (322) was reported in 2022, with more Ugandans than Kenyans compared to previous years, when there were more Kenyans (Fig 2). There was no identifiable seasonal variation/pattern in the number of cases diagnosed (Fig 3).

### Clinical characteristics of Visceral Leishmaniasis patients, Uganda, 2019–2024

Nine hundred and forty-three (97%) of the 972 VL patients were confirmed using the Leishmania dip-stick test. The most common symptoms of VL patients included fever (98%), night sweats (77%), abdominal swelling (72%), and weight loss

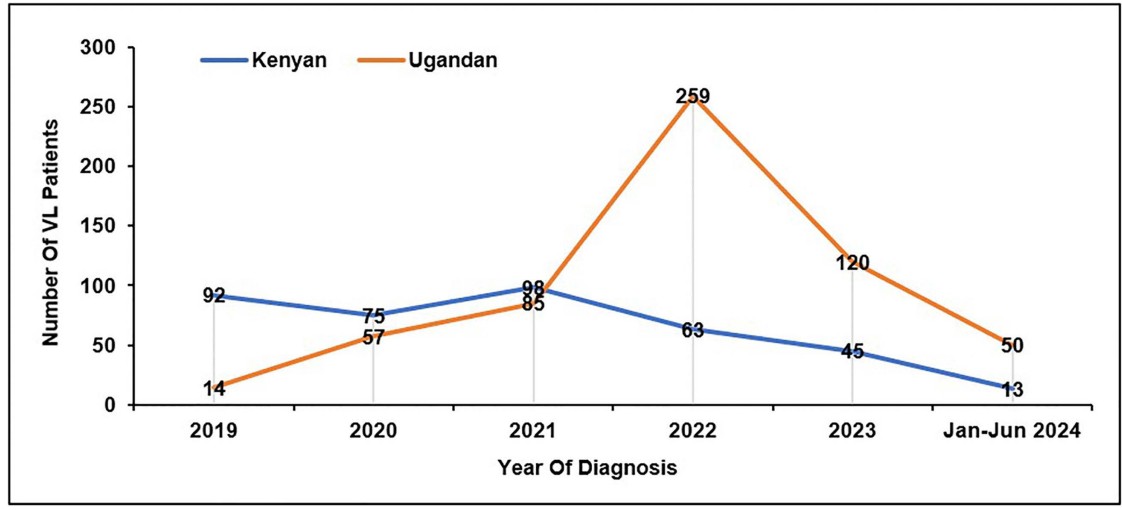

**Fig 2. Number of visceral leishmaniasis patients per year, Uganda, 2019–2024.** Annual distribution of confirmed VL cases treated at Amudat Hospital from 2019 to 2024, illustrating year-to-year variability and the peak observed in 2022.

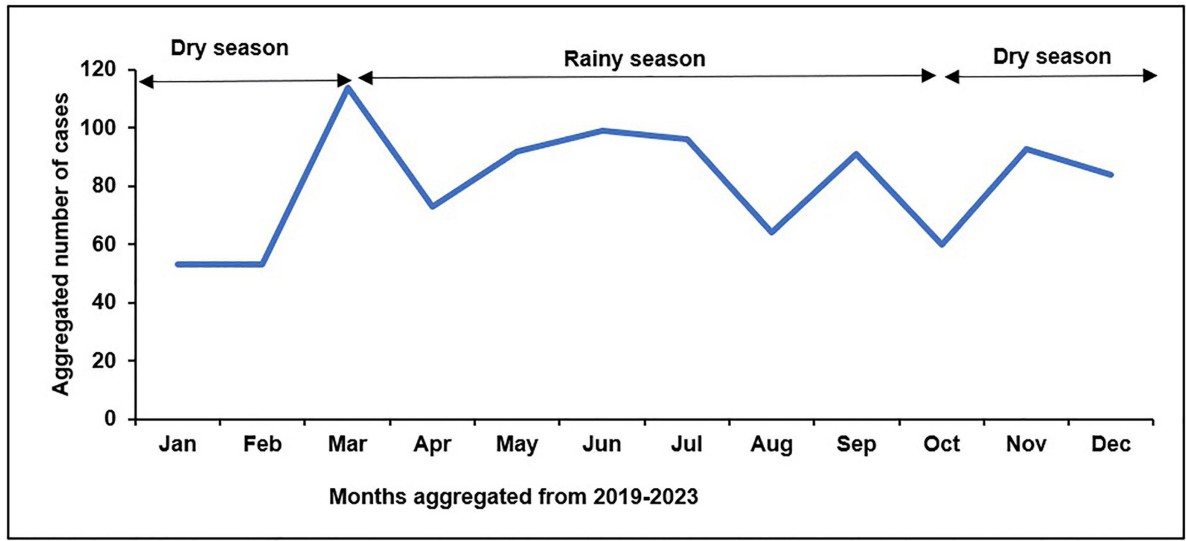

**Fig 3. Number of visceral leishmaniasis patients per month, Uganda, 2019–2023.** Monthly distribution of VL cases over the five-year period, showing seasonal patterns and fluctuations in patient presentation.

(68%). The average duration (SD) of sickness was 2.6 (0.3) months. Among patients tested for co-infections; 177 (18%) had malaria, 287 (30%) tested HIV positive, 2.7% had Hepatitis B. Additionally, 47% had severe anemia.

### Treatment characteristics and outcomes of Visceral Leishmaniasis patients, Uganda, 2019–2024

For most (895 [92%]) patients, the VL episode recorded was their index episode; only 8% were having a recurrence. Most patients received the first-line treatment regimen, 229 (24%) were treated with the second-line regimen, and almost all patients were cured; the overall case fatality rate was 1% (Table 1). Among patients on second-line treatment, 179 (78%) were HIV positive.

### Case fatalities and associated factors among Visceral Leishmaniasis patients, Uganda, 2019–2024

The CFR fluctuated between 0% and 2% across the years (1% in 2019, 2% in 2020 and 2021, 1% in 2022 and 0% in 2023 and 2024 (Fig 4). After adjusting for country of residence and Malaria, being HIV positive was the only measured predictor associated with death (Table 2).

## Discussion

Over five and a half years there were approximately 1,000 VL patients registered at the main treatment center in Amudat which translates to an average of approximately 55 patients a month. Most patients were children, male, and over one third were from neighboring Kenya predominantly the Turkana region, while Ugandan patients were predominantly from Moroto. There was no seasonal variation in VL observed and most cases were recorded in 2022. Most patients presented with fever, night sweats and abdominal swelling and for most the illness resolved in under three months. Malaria and HIV co-infection were common and many patients had severe anemia. The majority of HIV patients received second-line treatment regimen. Almost all patients completed treatment and were cured and the overall CFR was only approximately 1%, varying slightly across the years. Being HIV positive was associated with death.

Uganda and Kenya still account for 15 percent of the cases of Visceral leishmaniasis worldwide and it has been documented that cross-border migrations result in disease transmission between Uganda and Kenya [13]. This would explain why

**Table 1. Treatment and outcome characteristics of Visceral Leishmaniasis patients, Uganda, 2019–2024.**

| Treatment characteristics | Frequency | Percent |
| --- | --- | --- |
| **Illness episode** | | |
| Index episode | 898 | 92.4 |
| Recurrent episode | 74 | 7.6 |
| **Duration of treatment** | | |
| ≤ 2 weeks | 227 | 23.4 |
| > 2 | 745 | 76.6 |
| **Treatment regimen** | | |
| AmBisome (2nd line) | 229 | 23.6 |
| S.S.G/PM (1st line) | 743 | 76.4 |
| **Treatment outcome** | | |
| Completed/Failed | 1 | 0.1 |
| Completed/cured | 957 | 98.5 |
| Defaulted | 2 | 0.2 |
| Died | 12 | 1.2 |

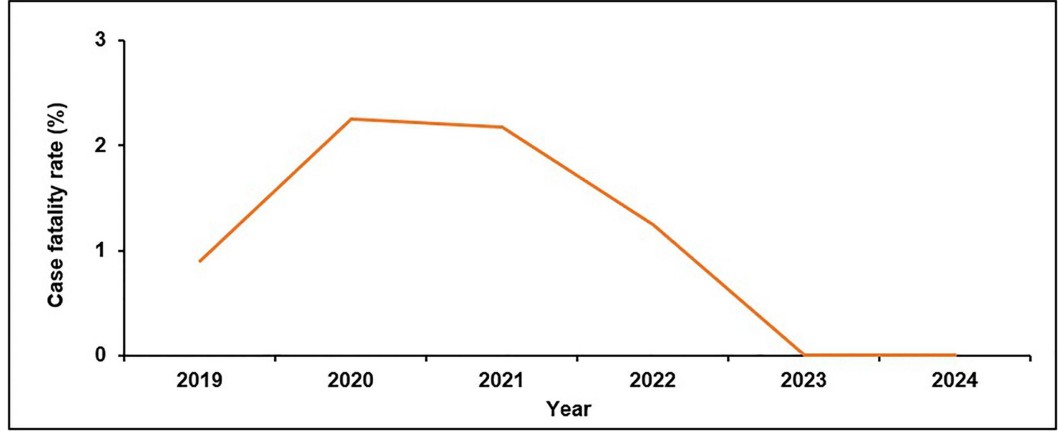

**Fig 4. Case fatality rate of visceral leishmaniasis patients per year, 2019–2024.** Annual case fatality rates among VL patients treated during the study period, highlighting variations in outcomes across the years.

more than one third of the VL patients in our study were from Kenya and some years even had more Kenyan patients than Ugandan. Similar studies in Uganda have reported more VL cases coming from Kenya highlighting the importance of Kenya in the persistence of VL in Uganda [15,28].It is also possible that some VL patients from Uganda are treated in Kenya which would mean that the VL counts for Uganda are an underestimate. However, this was not assessed in this study.

Furthermore, Ugandan patients were predominantly from Moroto while Kenyans were predominantly from Turkana County which is Kenya's largest county, directly bordering the Karamoja region. This relationship between the most endemic regions across the two countries which facilitates continuity of VL transmission highlights the fact that elimination of VL in Uganda hinges on elimination in Kenya. This presents an opportunity for targeted collaborative interventions such as enhanced active surveillance and consequent early treatment of cases; as well as enhanced vector control activities in the high-risk border districts between the two countries.

**Table 2.  Characteristics and factors associated with death among patients with Visceral Leishmaniasis, Uganda, 2019–2024.**

| Characteristics | Dead %(n/N) | Alive %(n/N) | OR | 95%CI | AOR | 95%CI |
|---|---|---|---|---|---|---|
| **Country/Residence** | | | | | | |
| Uganda | 0.7%(4/599) | 99%(595/599) | ref | | ref | |
| Kenya | 2%(8/373) | 98%(365/373) | 3.3 | 1.0-11.0 | 1.4 | 0.7-2.5 |
| **Age** | | | | | | |
| <5 years | 2%(2/129) | 98%(127/129) | ref | | | |
| 5-18 years | 0.3%(2/613) | 99%(611/613) | 0.2 | 0.03-1.5 | NA | |
| ≥19 years | 3.5%(8/230) | 97%(222/230) | 2.3 | 0.5-10.9 | NA | |
| **Sex** | | | | | | |
| Female | 0.7%(2/302) | 99%(300/302) | ref | | | |
| Male | 1.5%(10/670) | 98%(660/670) | 2.3 | 0.5-10.0 | NA | |
| **Severe Anemia** | | | | | | |
| No | 0.9%(4/460) | 99%(456/460) | | | | |
| Yes | 1.6%(8/512) | 98%(504/512) | 1.8 | 0.5-6.0 | NA | |
| **HIV status** | | | | | | |
| Negative | 0.5%(2/408) | 99(406/408) | | | | |
| Positive | 5.4%(10/185) | 95%(175/185) | 12 | 2.5-53.0 | **10** | **2.2-50.0** |
| **Malaria** | | | | | | |
| Negative | 1%(8/794) | 99%(786/794) | | | | |
| Positive | 2.3%(4/175) | 98%(171/175) | 2.3 | 0.7-8.0 | 3.4 | 0.97-12.0 |
| **Episode** | | | | | | |
| Index | 1.2%(11/898) | 99%(887/898) | | | | |
| Recurrent | 1.3%(1/74) | 99%(73/74) | 1.1 | 0.1-8.0 | NA | |

**OR: Odds ratio**

**AOR: Adjusted odds ratio**

**ref: Reference category**

**NA: not applicable, were not included in the multivariate model**

We found no seasonal variation in the number of VL cases over the years, based on the two climatic seasons experienced in the Karamoja Region of Uganda, which experiences only one rainy season from March to October [29]. This is similar to a 2019 report documenting no seasonal variation of VL in Uganda and Kenya [2]. However, in other non-African VL endemic countries, including India and Palestine, seasonal variations in VL incidence have been recorded. One study reviewing data for over 20 years in Palestine showed VL incidence peaking during the summer season and declining towards the winter season; while a study in India reported a peak in January which they attributed to variation in sand fly numbers likely to be positively correlated with temperature and negatively correlated with rainfall [30,31]. The difference between these findings and our study may be explained by the fact that since sandflies thrive under high temperatures, and Uganda/East Africa experiences high temperatures all-year round, it is likely that there is no significant variation in sand-fly populations across the seasons [10,32].

The peak of cases in 2022 could have been a post-COVID-19 rebound effect on health seeking behaviour as movement restrictions and service disruptions in the preceeding years could have temporarily suppressed care-seeking and reporting. However, this interpretation should be considered cautiously because we did not directly assess the influence of COVID-19 on VL service utilization.he commonest symptoms of VL included fever, weight loss, abdominal distension due to enlarged spleen and liver, and anemia [11]. All these symptoms were common among the patients in this study which likely supported the diagnosis in conjunction with the RDTs especially for the patients whose episode of VL was not their first.

Similar to malaria which is also another protozoan disease, studies have found that children are generally at greater risk of VL infection than adults in endemic areas and also suffer more complications such as severe anemia and death possibly due to their weaker immune systems [2,33–36]. In keeping with this, our study found that most VL patients were children and severe anemia was found in more than half of the cases. Just as it is with anemia of malaria, anemia of VL is caused by hemolysis due to alterations in erythrocyte membrane permeability, sequestration and destruction of the erythrocytes in the enlarged spleen and hemophagocytosis of infected red blood cells [37–40].

Our study revealed more male than female VL patients and although actual incidence was not determined, studies have reported higher risk and occurrence of VL among males, with females generally accounting for smaller proportions of VL cases [15,18,30]. This sex disparity is possibly due to the fact that males are more likely to engage in pastoral activities which increase their exposure to sandflies which transmit the Leishmania parasites. However, different from these findings, one study reviewing data for over 20 years in Palestine showed no sex difference in VL burden [31]; suggesting similar exposures across sexes to the sand fly in this region. This could indicate that in Palestinian pastoralist communities, both males and females participate in the herding activities.

Almost all patients were diagnosed using the leishmania dipstick test, which is an RDT recommended why WHO and allows for early diagnosis and treatment, and therefore an improved prognosis compared to previous parasitological tests, which were invasive and had a longer turnaround time [20]. However, being a serological test, it can remain positive after a past infection, although for most of the patients in this study, this was an index episode of VL. For patients with a past history of VL, this test is used in conjunction with a clinical finding pathognomonic of VL, including fever and abdominal distension, to prevent unnecessarily subjecting them to toxic drugs [1,20].

Co-infection with HIV was common in our study, in keeping with similar studies that have shown significant levels of VL-HIV co-occurrence. The significantly higher HIV prevalence among VL patients in Karamoja (30%) compared to the general population (2.1%) can be attributed to an interplay of biological, behavioral, and socio-cultural factors. HIV and VL are known to have a synergistic relationship—HIV weakens the immune system, making individuals more susceptible to Leishmania infection [41,42].

Additionally, risky sexual behavior often driven by poverty, conflict, or food insecurity—particularly among mobile and pastoralist communities common in Karamoja—may also contribute to HIV exposure. Many countries report cases, including Northern Ethiopia, which reportedly has the highest rate of HIV infection in VL patients, at 15–35% [41,43,44].

Our study found a similar co-infection rate to that of Ethiopia. VL is also included in WHO's clinical staging of HIV as an AIDS defining condition and the two infections reinforce each other, resulting in more severe illness, increasing the risk of death [18,42,45]. In keeping with this, our study showed that HIV infection was associated with death among VL patients. These findings present an opportunity for integrated programs targeting both HIV and VL in such populations. Co-infection with malaria was also observed in our study and has been documented in several studies with prevalences ranging from 4.2% to 30% [46–48].

Most patients in our study were treated with the first-line treatment regimen consisting of sodium stibogluconate plus paromomycin, although most HIV-coinfected patients were on the second-line regimen. This is in keeping with WHO and country guidelines for treatment of VL, which recommend second-line treatment for special categories including VL-HIV coinfection [20,45,49].

Mortality from VL in our study was low, almost reaching the WHO target for elimination as a public health problem [21]. This was lower than reported by previous studies (2009 and 2022) in Uganda, which reported annual CFRs as high as 11% [16,18]. This is indicative of how far the treatment and prognosis of VL have come over the years and the effect of effective treatment on reducing the burden of disease [22,50].

The findings from this study have important implications for other VL-affected countries in the Global South, many of which face similar challenges of cross-border transmission, fragile health systems, limited diagnostic capacity, and high burdens of co-infections such as HIV and malaria [51]. For instance, VL–HIV co-infection remains a major driver

of mortality, particularly in countries such as Ethiopia, where HIV-positive VL patients have significantly worse treatment outcomes and higher relapse rates [52].

Likewise, cross-border mobility has been documented as a key barrier to VL elimination. In East Africa, governments of Kenya, Ethiopia, Sudan, and other countries are increasingly prioritizing regional coordination for surveillance and treatment [51].In South Asia, persistent VL transmission in the India–Nepal–Bangladesh corridor, including new foci in border districts, highlights how population movement and cross-border infection risk complicate elimination efforts [53].

The high cure rates and low CFR in Amudat demonstrate the value of sustained availability of effective treatment, early diagnosis using RDTs, and programmatic emphasis on comorbidity management. These lessons are broadly applicable to countries with similar pastoralist or mobile populations, semi-arid ecology favorable to sandflies, and decentralized surveillance systems.

### Study limitations

The incidence of VL could not be determined as data were collected solely from Amudat Hospital and not from the newer treatment centers in Moroto and Napak. Consequently, the numbers reported likely underestimate the actual VL numbers in the country. Although Amudat Hospital is the main and longest-standing VL treatment center in Uganda, the study sample reflects a convenience sample rather than a nationally representative dataset. During most of the study period, Amudat functioned as the primary center for VL diagnosis while the other centers only became operational in the last two years. As a result, the findings predominantly represent patients who sought care at Amudat and may not capture VL patterns in districts where individuals accessed services across the border in Kenya or at the newer Ugandan treatment centers."

Also, given that the burden of disease was estimated passively based on existing hospital records, it might be an underestimation of actual burden considering the likelihood of subpar health seeking behaviour in the region.

Accurate determination of prevalence of co-infections was precluded by the fact that not all patients were tested for specific co-morbidities. For example, only 28 patients had results for Hepatitis B among whom 26 were positive, which is not an accurate representation of the prevalence of Hepatitis B among all VL patients.

## Conclusion

Our findings reveal high cure rates and low case fatality rates among patients treated for VL in Uganda, suggesting meaningful progress towards elimination of VL as a public health problem. We also highlight the significance of the cross-border relationship between Uganda and Kenya in the epidemiology of disease and also underscore the prognostic importance of co-infections especially HIV.

However, these findings should be interpreted with caution because the sample used reflects patients treated at Amudat Hospital,but does not fully represent all VL cases nationally, particularly those managed at the newer centers in Moroto and Napak or across the border in Kenya, which limits generalizability.

Despite these limitations, the study provides critical patient-level evidence that can guide targeted interventions, inform VL program strengthening, and support ongoing efforts toward VL elimination in Uganda and the broader East African region."

### Recommendations

We recommend targeted interventions—such as enhanced active surveillance and consequent early treatment of cases, as well as enhanced vector control activities in the high-risk border districts between Kenya and Uganda—to address cross-border vector and human-to-human transmission, as well as routine prompt screening/treatment of co-infections.

In addition, strengthening border-region surveillance, harmonizing treatment protocols across neighboring countries, and integrating VL services within HIV and malaria programs may therefore accelerate progress toward VL elimination in Uganda and across multiple regions of the Global South.

## Acknowledgments

The authors thank the Vector Borne and Neglected Tropical Diseases Division of the Ministry of Health for their technical guidance.

## Author contributions

**Conceptualization:** Benigna Gabriela Namara, Ivan Ankunda, Benon Kwesiga, Lilian Bulage, Sandra Nabatanzi, Alex Riolexus Ario, Alfred Mubangizi, Daniel Kadobera.

**Data curation:** Benigna Gabriela Namara, Richard Migisha, Daniel Kadobera.

**Formal analysis:** Benigna Gabriela Namara, Richard Migisha.

**Funding acquisition:** Benigna Gabriela Namara, Richard Migisha, Alex Riolexus Ario, Daniel Kadobera.

**Investigation:** Benigna Gabriela Namara, Ivan Ankunda.

**Methodology:** Benigna Gabriela Namara, Ivan Ankunda, Daniel Kadobera.

**Project administration:** Benigna Gabriela Namara, Ivan Ankunda, Alex Riolexus Ario.

**Resources:** Benigna Gabriela Namara, Alex Riolexus Ario, Daniel Kadobera.

**Software:** Benigna Gabriela Namara, Alex Riolexus Ario.

**Supervision:** Benigna Gabriela Namara, Ivan Ankunda, Richard Migisha, Daniel Kadobera.

**Validation:** Benigna Gabriela Namara, Alfred Mubangizi, Daniel Kadobera.

**Visualization:** Benigna Gabriela Namara.

**Writing – original draft:** Benigna Gabriela Namara, Lilian Bulage.

**Writing – review & editing:** Benigna Gabriela Namara, Richard Migisha, Benon Kwesiga, Lilian Bulage, Sandra Nabatanzi, Alex Riolexus Ario, Alfred Mubangizi, Daniel Kadobera.

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
