## [Decision Letter · Decision Letter 0]

23 Oct 2025

Characteristics, Treatment Outcomes and Factors Associated with Death among Patients with Visceral Leishmaniasis, Uganda, 2019–2024

Dear Dr. Namara,

Thank you for submitting your manuscript to PLOS Neglected Tropical Diseases. After careful consideration, we feel that it has merit but does not fully meet PLOS Neglected Tropical Diseases's publication criteria as it currently stands. Therefore, we invite you to submit a revised version of the manuscript that addresses the points raised during the review process.

Please submit your revised manuscript within 60 days Dec 22 2025 11:59PM. If you will need more time than this to complete your revisions, please reply to this message or contact the journal office at plosntds@plos.org. Please include the following items when submitting your revised manuscript:

We look forward to receiving your revised manuscript.

Kind regards,

Angamuthu Selvapandiyan, Ph.D.

Academic Editor

Susan Madison-Antenucci

Section Editor

Shaden Kamhawi

co-Editor-in-Chief

Paul Brindley

co-Editor-in-Chief

**Additional Editor Comments:**

The authors must describe the study design in the methods section. The figures in the results needs resolution (Figure 1) and for the rest of the figures, the legends/labbels in the figures must have uniformuty with text font and theirs sizes throughout.

**Journal Requirements:**

At this stage, the following Authors/Authors require contributions: Benigna Gabriela Namara, Ivan Ankunda, Richard Migisha, Benon Kwesiga, Lilian Bulage, Sandra Nabatanzi, Alex Riolexus Ario, Alfred Riolexus Mubangizi, and Daniel Riolexus Kadobera. Please ensure that the full contributions of each author are acknowledged in the "Add/Edit/Remove Authors" section of our submission form.

3) Some material included in your submission may be copyrighted. According to PLOSu2019s copyright policy, authors who use figures or other material (e.g., graphics, clipart, maps) from another author or copyright holder must demonstrate or obtain permission to publish this material under the Creative Commons Attribution 4.0 International (CC BY 4.0) License used by PLOS journals. Please closely review the details of PLOSu2019s copyright requirements here: PLOS Licenses and Copyright. If you need to request permissions from a copyright holder, you may use PLOS's Copyright Content Permission form.

Potential Copyright Issues:

- Figure 1. Please (a) provide a direct link to the base layer of the map (i.e., the country or region border shape) and ensure this is also included in the figure legend; and (b) provide a link to the terms of use / license information for the base layer image or shapefile. We cannot publish proprietary or copyrighted maps (e.g. Google Maps, Mapquest) and the terms of use for your map base layer must be compatible with our CC BY 4.0 license.

4) In the online submission form, you indicated that "The datasets upon which our findings are based belong to the Uganda Public Health Fellowship Program (PHFP) Ministry of Health and are not publicly available,but can be availed upon reasonable request from the corresponding author with permission from the Uganda Public Health Fellowship Program.". All PLOS journals now require all data underlying the findings described in their manuscript to be freely available to other researchers, either

1. In a public repository

2. Within the manuscript itself

3. Uploaded as supplementary information.

State the initials, alongside each funding source, of each author to receive each grant. For example: "This work was supported by the National Institutes of Health (####### to AM; ###### to CJ) and the National Science Foundation (###### to AM).".

**Reviewers' Comments:**

Reviewer's Responses to Questions

**Key Review Criteria Required for Acceptance?**

**Methods:**

-Are the objectives of the study clearly articulated with a clear testable hypothesis stated?

-Is the study design appropriate to address the stated objectives?

-Is the population clearly described and appropriate for the hypothesis being tested?

-Is the sample size sufficient to ensure adequate power to address the hypothesis being tested?

-Were correct statistical analysis used to support conclusions?

-Are there concerns about ethical or regulatory requirements being met?

Reviewer #1: Need for study (Background) needs to be specified more clearly.

Methodology is clear.

Reviewer #2: #Reviewer Comments

I appreciate the opportunity to review this manuscript, which addresses a topic of significant relevance to Uganda and the African continent. Below I provide specific considerations and suggestions that may strengthen the work.

#Abstract

- It may be more effective to begin by reaffirming that “visceral leishmaniasis, a neglected tropical disease, continues to affect several countries, including Uganda.”

- Consider highlighting the underlying reasons for the persistence of this disease globally and in East Africa.

- In the Methods section, it is important to specify the study design (which appears to be observational and cross-sectional). This should also be made explicit in the main text of the manuscript.

- In the Results section, please review the expression “(80%, [259/322).” The presentation of these results should be standardized throughout the manuscript.

#Theoretical Foundation

- It would strengthen the contextualization to cite elimination programs conducted in East Africa and in India, as these provide relevant comparative experiences.

#Methodology

- Figure 1 is relevant and informative, but a higher-resolution version would be beneficial.

- In Figure 2, consider harmonizing the font used for the numbers with that of the main text for consistency.

- The study is limited to a single center (Amudat Hospital), which reduces its national representativeness. It is important to clarify whether Amudat is the only treatment center or whether this is a convenience sample.

- Please provide a stronger justification for the choice of study period and for the exclusion of other reference centers. Was this due to operational constraints (e.g., data availability), or were there other specific reasons?

- The manuscript does not mention approval by a Human Research Ethics Committee. This is a critical omission, particularly given the use of patient medical records. Information about ethical clearance and patient confidentiality protections must be included.

**Results**

-Does the analysis presented match the analysis plan?

-Are the results clearly and completely presented?

-Are the figures (Tables, Images) of sufficient quality for clarity?

Reviewer #1: No comments for authors.

Reviewer #2: #Results and Discussion

- The results are well described and supported by clear quantitative evidence;

- The statement about a “post-COVID-19 rebound effect” should be qualified as a possibility rather than presented as fact, unless supporting references are provided;

- Some paragraphs are too long (lines 265-297; 331-348), making them difficult to read;

- A valuable addition would be to expand the discussion on how lessons learned from this study could inform strategies in other countries of the Global South facing similar challenges. Making comparisons with other countries in the Global South would elevate the work to a broader level and context.

**Conclusions:**

-Are the conclusions supported by the data presented?

-Are the limitations of analysis clearly described?

-Do the authors discuss how these data can be helpful to advance our understanding of the topic under study?

-Is public health relevance addressed?

Reviewer #1: No comments for authors.

Reviewer #2: #Conclusions

The authors could expand their conclusions by including considerations for interpreting the results. However, the sample used in the analysis needs improvement.

**Editorial and Data Presentation Modifications?**

Reviewer #1: Minor Revision

Table 1 and Introduction.

Reviewer #2: #General comment

The manuscript addresses an important public health issue and presents useful findings. Incorporating the clarifications above (particularly regarding methodology, ethical approval, and contextual discussion) would improve the rigor and impact of the work.

**Summary and General Comments:**

Reviewer #1: PEER REVIEW REPORT PNTD-D-25-01017

Summary

This retrospective study examines Visceral Leishmaniasis cases in Uganda from 2019 to 2024, analysing patient characteristics, treatment outcomes, and mortality factors to evaluate progress toward World Health Organization elimination targets. Visceral Leishmaniasis, commonly known as kala-azar, represents a significant neglected tropical disease burden in Uganda's semi-arid Karamoja region, which borders Kenya where the disease is endemic. The research utilized secondary data from patient files at the primary treatment facility in Amudat District, employing logistic regression analysis to identify mortality-associated factors.

The analysis encompassed 972 patient records, revealing that most patients were male (69%) and under 18 years of age (76%). Notably, 38% of cases originated from neighbouring Kenya, while 72% of Ugandan patients came from Moroto District. Case numbers peaked at 322 in 2022, with a marked shift toward more Ugandan patients compared to earlier years when Kenyan cases predominated. Clinical manifestations commonly included fever (98%), night sweats (77%), and abdominal swelling (72%). Severe anaemia affected 53% of patients, while co-infections were frequent, with malaria present in 18% and HIV in 31% of cases.

Treatment outcomes demonstrated remarkable success, with 98% of patients achieving cure rates using predominantly first-line therapy regimens. The overall case fatality rate remained low at 1%, declining from 2% in 2020-2021 to below 1% in 2023-2024. HIV co-infection emerged as the strongest mortality predictor, with an adjusted odds ratio of 10. These findings indicate substantial progress toward elimination goals through effective treatment protocols and high cure rates. However, persistent cross-border transmission challenges and the critical impact of HIV co-infection underscore the need for enhanced surveillance, early intervention strategies, and integrated public health approaches targeting high-risk border areas to achieve complete elimination objectives.

Remarks

Great work Authors.

I am really sorry for my late review.

Introduction- Towards the end of the background, the motive behind this study should be specifically mentioned (Need for study/ Rationale).

Table 1- Treatment outcome percentages are not totaling to 100.

Thank You

Reviewer #2: The study has a significant methodological problem that impacts the extrapolation of results. Furthermore, the manuscript fails to mention approval by a Human Research Ethics Committee.

PLOS authors have the option to publish the peer review history of their article (what does this mean? ). If published, this will include your full peer review and any attached files.

**Do you want your identity to be public for this peer review?** For information about this choice, including consent withdrawal, please see our Privacy Policy .

Reviewer #1: **Yes:** DR. DENNY MATHEW JOHN

Department of Community Medicine

Kerala Medical College Hospital

Palakkad, Kerala, India

Reviewer #2: No

**Figure resubmission:**
---

## [Decision Letter · Decision Letter 1]

12 Feb 2026

Dear Namara,

We are pleased to inform you that your manuscript 'Characteristics, Treatment Outcomes and Factors Associated with Death among Patients with Visceral Leishmaniasis, Uganda, 2019–2024' has been provisionally accepted for publication in PLOS Neglected Tropical Diseases.

Best regards,

Susan Madison-Antenucci

Section Editor

Shaden Kamhawi

co-Editor-in-Chief

Paul Brindley

co-Editor-in-Chief

Reviewer's Responses to Questions

**Key Review Criteria Required for Acceptance?**

**Methods**

-Are the objectives of the study clearly articulated with a clear testable hypothesis stated?

-Is the study design appropriate to address the stated objectives?

-Is the population clearly described and appropriate for the hypothesis being tested?

-Is the sample size sufficient to ensure adequate power to address the hypothesis being tested?

-Were correct statistical analysis used to support conclusions?

-Are there concerns about ethical or regulatory requirements being met?

Reviewer #1: All necessary changes made.

Reviewer #2: #Reviewer Comments

I appreciate the opportunity to review this manuscript, which addresses a topic of great relevance to Uganda and the African continent. As this is a new submission, it is understood that the authors have truly made an effort to refine the work.

Below, I present specific considerations and suggestions that may improve the work.

>Regarding the quality of Figure 1:

The map has improved significantly, but it could still have better resolution.

>The authors clarified the presentation of the data “(80%, [259/322).” Lines 38-41.

#Methods

-Objectives and methods are well established. The sample size is a significant limitation. However, it should be understood that this is a difficult reality. In this context, it should be considered that this is the available information.

>Regarding the study design:

Minimum information about the study design has been entered. OK.

>Regarding Amudat Hospital;

"Convenience sampling" has little evidentiary power. If possible, include the percentage of the hospital's patient volume in relation to the country. It is important for the reader to understand that this is the main hospital responsible for treating these cases. This can enhance the analysis. It should also be mentioned for consideration in the conclusion.

>Regarding the study period;

OK

>Regarding approval by a Research Ethics Committee involving Human Subjects;

Plos One has a specific section to include this information.

**Results**

-Does the analysis presented match the analysis plan?

-Are the results clearly and completely presented?

-Are the figures (Tables, Images) of sufficient quality for clarity?

Reviewer #1: Necessary modifications done.

Reviewer #2: 2#Results

-The results are understandable. Although Figure 1 has improved, the resolution could still be improved to meet Plos One's quality standards.

>Regarding the "post-COVID-19 rebound effect";

OK. But check line 294 ("service utilization.the commonest").

**Conclusions**

-Are the conclusions supported by the data presented?

-Are the limitations of analysis clearly described?

-Do the authors discuss how these data can be helpful to advance our understanding of the topic under study?

-Is public health relevance addressed?

Reviewer #1: Yes.

Reviewer #2: 3#Conclusions

-Despite the limitations of the sample, the results are useful for understanding the national context.

**Editorial and Data Presentation Modifications?**

Reviewer #1: Accept.

Reviewer #2: 4#Editorial and Data Presentation Modifications?

-The work was adjusted.

**Summary and General Comments**

Reviewer #1: Great work authors.

Sorry for the late reply.

All necessary changes have been duly made.

Reviewer #2: 5#Summary and General Comments

-The work has improved significantly.

PLOS authors have the option to publish the peer review history of their article (what does this mean? ). If published, this will include your full peer review and any attached files.

**Do you want your identity to be public for this peer review?** For information about this choice, including consent withdrawal, please see our Privacy Policy .

Reviewer #1: **Yes:** DR. DENNY MATHEW JOHN

Department of Community Medicine

Kerala Medical College Hospital

Palakkad, Kerala, India

Reviewer #2: No

---

## [Editor Report · Acceptance letter]

Dear Namara,

We are delighted to inform you that your manuscript, "Characteristics, Treatment Outcomes and Factors Associated with Death among Patients with Visceral Leishmaniasis, Uganda, 2019–2024," has been formally accepted for publication in PLOS Neglected Tropical Diseases.

Best regards,

Shaden Kamhawi

co-Editor-in-Chief

Paul Brindley

co-Editor-in-Chief
